# Calpain Regulates Reactive Oxygen Species Production during Capacitation through the Activation of NOX2 and NOX4

**DOI:** 10.3390/ijms24043980

**Published:** 2023-02-16

**Authors:** César I. Ortiz-García, Monica L. Salgado-Lucio, Ana L. Roa-Espitia, Aidé A. Muñoz-Sánchez, Joaquín Cordero-Martínez, Enrique O. Hernández-González

**Affiliations:** 1Department of Cell Biology, Centro de Investigación y Estudios Avanzados del Instituto Politécnico Nacional, Av. Instituto Politécnico Nacional 2508, San Pedro Zacatenco, Del. Gustavo A. Madero, México City 07360, Mexico; 2Department of Health Sciences, Universidad Autónoma Metropolitana, División de Ciencias Biológicas y de la Salud, Unidad Iztapalapa, Av. San Rafael Atlixco No. 186, Colonia Vicentina, Alcaldía Iztapalapa, México City 09310, Mexico; 3Department of Biochemistry, Escuela Nacional de Ciencias Biológicas, Instituto Politécnico Nacional, Prolongación Manuel Carpio y Plan de Ayala s/n Col, Santo Tomás, Del. Miguel Hidalgo, México City 11340, Mexico

**Keywords:** NADPH oxidases, ROS production, capacitation, acrosome reaction, mobility, calcium, calpain

## Abstract

Capacitation is a series of physiological, biochemical, and metabolic changes experienced by mammalian spermatozoa. These changes enable them to fertilize eggs. The capacitation prepares the spermatozoa to undergo the acrosomal reaction and hyperactivated motility. Several mechanisms that regulate capacitation are known, although they have not been fully disclosed; among them, reactive oxygen species (ROS) play an essential role in the normal development of capacitation. NADPH oxidases (NOXs) are a family of enzymes responsible for ROS production. Although their presence in mammalian sperm is known, little is known about their participation in sperm physiology. This work aimed to identify the NOXs related to the production of ROS in guinea pig and mouse spermatozoa and define their participation in capacitation, acrosomal reaction, and motility. Additionally, a mechanism for NOXs’ activation during capacitation was established. The results show that guinea pig and mouse spermatozoa express NOX2 and NOX4, which initiate ROS production during capacitation. NOXs inhibition by VAS2870 led to an early increase in the capacitation and intracellular concentration of Ca^2+^ in such a way that the spermatozoa also presented an early acrosome reaction. In addition, the inhibition of NOX2 and NOX4 reduced progressive motility and hyperactive motility. NOX2 and NOX4 were found to interact with each other prior to capacitation. This interaction was interrupted during capacitation and correlated with the increase in ROS. Interestingly, the association between NOX2-NOX4 and their activation depends on calpain activation, since the inhibition of this Ca^2+^-dependent protease prevents NOX2-NOX4 from dissociating and ROS production. The results indicate that NOX2 and NOX4 could be the most important ROS producers during guinea pig and mouse sperm capacitation and that their activation depends on calpain.

## 1. Introduction

In vivo reactive oxygen species (ROS) are a secondary product in many cellular processes, such as the electron transport chain, lipid peroxidation, and the xanthine oxidase system. The traditional views of ROS only address their roles related to aging and disease. However, in recent years, they have gained attention as signaling molecules in many physiological processes, such as cell differentiation, vascular pressure regulation, and sperm physiological maturation, i.e., capacitation and hyperactivation [1].

ROS include dioxygen (O_2_) molecules reduced by one (radical) or two (nonradical) electrons, and these are classified by their reduction potential. Superoxide anion (O_2_^•^-) and hydrogen peroxide (H_2_O_2_) are ROS that have the potential to serve as second messengers because cells have specialized systems for ROS production, such as the NADPH oxidase (NOX) system [1,2]. NOXs catalyze ROS production by transferring electrons from NADPH to O_2_ in a highly regulated manner. NOXs’ function and regulation were first described in phagocytic cells. The phagocytic ROS production system Nox2/gp91^phox^ is composed of several subunits: the catalytic and transmembrane subunit flavocytochrome b588 integrated by Nox2/gp91^phox^ and p22^phox^, and the cytosolic and regulatory proteins p47^phox^, p67^phox^, as well as the small GTPase, Rac1 [2,3]. 

In mammalian sperm processes such as capacitation, acrosome reaction and motility are sensitive to ROS such as O_2_^•^-, H_2_O_2_, and nitric oxide (NO) [4,5]. Human spermatozoa were the first cells in which physiological ROS production was reported [6]. The detection of NOX5, whose activity is calcium-dependent, is responsible for the oxidative stress produced during capacitation and regulates capacitation and motility in human spermatozoa [7]. However, in other mammals, such as rodents (i.e., mice, rats, and guinea pigs), the *nox5* gene is absent from their genomes [2]. Even so, ROS production has been reported in mouse and rat spermatozoa. Members of the NOX family, NOX2 and NOX3, respectively, have been observed in sperm cells, which suggests that ROS production by NOX family members plays a significant role in physiological processes, such as capacitation, acrosome reaction, and mobility development [7,8,9]. Interestingly, the NOX1 and NOX2 activator p67phox is a Rac1 target, so Rac is essential to regulate NOX1 and NOX2 activity [2,3].

ROS production through NADPH oxidases has been correlated with the activation of calcium-dependent proteases and calpains [10,11]. Calpain activity is essential for fertilization [12], since it allows mammalian spermatozoa to acquire their fertilizing ability by participating in physiological processes such as capacitation, acrosomal reaction, and sperm mobility [13,14,15,16,17], as well as in fowl sperm mobility [18]. Calpains-1 and -2 are the two isoforms of the calpain family members found in human, mouse, and guinea pig spermatozoa [12,14,17]. Although the mechanism by which these proteases regulate capacitation, acrosomal reaction, and mobility is not well understood, calpain-1 is known to regulate remodeling of the spectrin cytoskeleton [14], lipid raft rearrangement, and activation of the Src kinase family [15].

Given the existence of multiple members of the NOX family in mammal cells, and the presence of calpains [13,14,15,16,17] and Rac1 [19,20] in mammalian spermatozoa, as well as the importance of ROS in several sperm processes that prepare them for fertilization, such as capacitation, acrosomal reaction, and mobility [21], the objective of this work was to determine which members of the NOX family are responsible for the production of ROS in guinea pig spermatozoa. Furthermore, we examined the participation of ROS produced by NOX in capacitation, acrosomal reaction, and mobility, and whether the activity of the NOX present in guinea pig spermatozoa is dependent on calpain. The results showed that, unlike human and stallion spermatozoa [7,22], two different NOX members are present in guinea pig spermatozoa. We also found that the production of ROS by these NOXs is related to capacitation, acrosomal integrity, and mobility. The results also suggested that NOX2 and NOX4 interact in the non-capacitated state but not in the capacitated state. Such interactions and NOX activity are dependent on calpain activity.

## 2. Results

### 2.1. Guinea Pig and Mouse Spermatozoa Express Two Different Flavoproteins Implicated in NADPH Consumption

The correct capacitation in many mammalian spermatozoa is strongly related to ROS production. First, our work focused on determining which NOX family proteins are involved in ROS production and function during capacitation and the role that ROS play in guinea pig spermatozoa. In this way, we quantified the specific NADPH consumption (µmol/min/mg-protein) in native total protein extracts in the presence of ROS production inhibitors at different levels using diphenyl iodonium (DPI), a general flavoprotein inhibitor, and the Rac1 inhibitor NSC23766, which prevents Rac1 activation and is necessary for the activation of Nox1 and 2. This assay revealed that NADPH consumption in native total extracts was significantly increased under conditions of capacitation due to a flavoprotein system, while DPI abolished NADPH consumption (Figure 1A). The Rac1 inhibitor (NSC23766) was unable to prevent NADPH consumption; conversely, the consumption levels of NADPH were higher, although not significant, compared to those shown in the extracts of capacitated spermatozoa, a rise that was inhibited by DPI (Figure 1A). These results suggest the presence of a flavoprotein system composed of at least two different flavoproteins, which consume NADPH in spermatic native total extracts. Rac1 regulates one of the NOXs (1 or 2), while the other would show independent Rac1 activity.

We hypothesize that NOX2 and NOX4 are the two NOX that are related to ROS production in guinea pig spermatozoa. The reasons for this suggestion are as follows: NOX2 has been found in rodents’ spermatozoa, such as that of mice and rats; NOX4 produces constitutively large amounts of peroxide and is not regulated by Rac1. This hypothesis was confirmed by Wb: using specific antibodies that recognize NOX2 and NOX4, we detected protein bands of approximately 65 kDa for both NOX2 and NOX4 in guinea pig and mouse spermatozoa (Figure 1B). Similar protein bands were detected in the testicles and kidneys of guinea pigs and mice (Figure 1C).

### 2.2. DPI Affects Sperm Viability

With the aim of learning the effects of NOX antagonists on sperm viability, this was evaluated by PI assay. The test results show no significant differences between non-capacitated, capacitated, and capacitated sperm in the presence of VAS2870, NSC23766, or ML171 (Figure 2A). However, DPI clearly decreased sperm viability with respect to those not treated with DPI (Figure 2A). For this reason, this NOX inhibitor was not used in tests carried out in vivo.

### 2.3. NADPH Oxidase Inhibition Abolishes ROS Production during Capacitation

To determine whether ROS (O_2_^•^- and H_2_O_2_) are produced during capacitation by NOXs, spermatozoa were capacitated, and H_2_O_2_ and O_2_^•^- were evaluated. The results showed that spermatozoa in a non-capacitated state produce a basal level of O_2_^•^- and H_2_O_2_ and, after 60 min of capacitation, the levels of O_2_^•^- and H_2_O_2_ significantly increased (Figure 2B,C). To define whether the increases in O_2_^•^- and H_2_O_2_ are a consequence of NOX activity, VAS2870, a pan-NOX inhibitor, and NSC23766, a specific Rac1 inhibitor, were tested. The O_2_^•^- increase experienced by spermatozoa during capacitation was inhibited by VAS2870 and NSC23766, maintaining similar levels to those observed in non-capacitated spermatozoa (Figure 2B). These results suggest that NOX2 is active in guinea pig spermatozoa. In the case of H_2_O_2_, the increase experienced by spermatozoa during capacitation was significantly inhibited when spermatozoa were capacitated in the presence of VAS2870, while NSC23766 did not inhibit H_2_O_2_ production, suggesting that NOX4 remains active (Figure 2C). To confirm that this increase in H_2_O_2_ production is the consequence of NOX4 activity, sperm capacitation was performed in the presence of ML171 (2 µM), a more specific inhibitor for NOX4. ML171 significantly inhibited H_2_O_2_ production regarding the spermatozoa that were capacitated in the absence of ML171 (Figure 2C); however, H_2_O_2_ production in the presence of ML1171 is significantly (*p* < 0.01) higher when considering non-capacitated spermatozoa (Figure 2C). Possibly, this H_2_O_2_ is produced by NOX2, which is not affected by ML171. NSC23766 and ML171 were tested during capacitation, finding a reduction in H_2_O_2_ to levels that were very similar to the non-capacitated spermatozoa (Figure 2C). These did not show a significant difference. Together, these results indicate that NOX2 and NOX4 are the main producers of ROS in mammalian spermatozoa.

### 2.4. The Effects of Inhibiting ROS Production on Sperm Physiology

During capacitation, the levels of different ROS rose, exerting a positive effect on capacitation. Interfering with the generation of these ROS impairs capacitation and other sperm processes, such as the acrosomal reaction and hypermotility. To define whether ROS generation by NOX2 and NOX4 in guinea pig spermatozoa is essential for capacitation, spontaneous acrosomal reaction (sAR) and motility, they were valued in spermatozoa capacitated in the presence of VAS2870 (40 µM).

#### 2.4.1. Capacitation

The evaluation using the CTC assay showed that, under conditions of capacitation, pattern B slowly increased during the first minutes of incubation. After 15 min of incubation, the percentage of spermatozoa showing pattern B rapidly increased, reaching a maximum at 30 min of incubation (Figure 3A). After 60 min of capacitation, the rate of pattern B decreased, probably as a consequence of the increase in the number of spermatozoa that showed pattern AR (Figure 3B). Unlike the control, spermatozoa capacitated in the presence of VAS2870 showed a rapid rise in pattern B, reaching its maximum at 15 min of incubation. The percentage of pattern B rapidly decreased (Figure 3A), perhaps due to the rapid increase in the AR pattern (Figure 3B). These results suggest that the inhibition of NOX2 and NOX4 accelerates capacitation.

The effect of the inhibition of ROS production on capacitation is usually assessed using the protein tyrosine phosphorylation assay (PYP). The results show that the inhibition of ROS generation during capacitation prevents PYP. To determine whether PYP occurs during NOX inhibition, PYP was valued. Thus, spermatozoa capacitated in the presence of VAS2870 (40 µM) showed PYP levels similar to non-capacitated spermatozoa, which are significantly lower than those presented by spermatozoa capacitated in the absence of VAS2870 (Figure 3C,D). These results confirm that inhibiting ROS generation during capacitation prevents PYP.

#### 2.4.2. Spontaneous Acrosomal Reaction

CTC evaluation shows that, in spermatozoa capacitated in the presence of VAS2870, the pattern AR rapidly increased, reaching significantly higher levels than those capacitated in the absence of VAS2870 (Figure 3B). This result may be a consequence of the rapid increase in pattern B that occurred during capacitation in the presence of VAS2870.

#### 2.4.3. Motility

To determine the effects of H_2_O_2_ and O_2_^•^- on motility, total motility and progressive motility were assessed in capacitated spermatozoa in the presence of VAS2870. The results revealed a significant increase in both parameters in capacitated compared to non-capacitated spermatozoa (Figure 4A,B). The total and progressive motility in the presence of VAS2870 significantly inhibited this increase (Figure 4A,B).

In summary, these results demonstrated that, during capacitation, the inhibition of O_2_^•^- and H_2_O_2_ production by VAS2870 significantly affects the physiology of guinea pig spermatozoa.

### 2.5. ROS Reduction by NOX Inhibition Increases Intracellular Ca^2+^

The PYP results suggest that capacitation is inhibited when sperm are incubated in the presence of VAS2870. Despite this, sAR is increased early (Figure 3B), suggesting that calcium influx is not altered by NOX inhibition. To confirm this hypothesis, the [Ca^2+^]i during capacitation was assessed in the presence or absence of VAS2870 (40 µM). The results show that the [Ca^2+^]i gradually increased in capacitated spermatozoa. It reached its maximum at 30 min of incubation and remained at these levels until 60 min of capacitation (Figure 5). After 60 min, the [Ca^2+^]i rapidly declined due to sAR, while the [Ca^2+^]i quickly increased in spermatozoa capacitated in the VAS2870 presence, reaching a maximum at 30 min of incubation and rapidly decaying after this time (Figure 5). [Ca^2+^]i at 15 and 30 min of capacitation in the presence of VAS2870 is significantly higher than the [Ca^2+^]i in the absence of VAS2870 (Figure 5). Capacitated spermatozoa in the presence of VAS2870 may experience early capacitation and sAR.

### 2.6. NOX2 and NOX4 Alter Their Subcellular Localization during Capacitation

To determine whether NOX2 and NOX4 exhibit differential subcellular localization, both proteins were immunolocalized in non-capacitated and capacitated spermatozoa. NOX2 and NOX4 were scattered throughout the acrosome and in the midpiece of the non-capacitated spermatozoa (Figure 1D). During capacitation, the NOX2 and NOX4 localized in the acrosome aggregated in the apical region (Figure 6A, middle panels). Since a relationship has been suggested between NOX2 and the Ca^2+^-dependent protease calpain, and guinea pig spermatozoa express calpain 1, we investigated whether calpain activity was correlated with NOX aggregation in the apical region of the acrosome. To corroborate this hypothesis, spermatozoa were capacitated in the presence of a specific calpain inhibitor, calpeptin (10 µM). The localization analysis of both NOXs showed that the inhibition of calpain blocked the aggregation of NOX2 and NOX4 (Figure 6A, right panels). Their localization was similar to that of the non-capacitated spermatozoa. Changes in NOX2 and NOX4 were not observed in the midpiece of the flagellum (Figure 6A).

### 2.7. The Association between NOX2 and NOX4 Depends on Calpain

Since both NOXs aggregate at the same acrosome site, they could have a direct or indirect physical interaction. To test this hypothesis, we examined whether NOX2 coimmunoprecipitates with NOX4. The results showed that NOX2 coimmunoprecipitated with NOX4 in non-capacitated spermatozoa; however, this did not occur in capacitated spermatozoa (Figure 6B,C). These results indicate that NOX2 and NOX4 are associated (NOX2-NOX4) in non-capacitated spermatozoa and lose this association during capacitation. When spermatozoa were capacitated in the presence of VAS2870, NOXs’ dissociation was not interrupted (Figure 6B,C), suggesting that NOX2 and NOX4 dissociation does not depend on their activation but on a factor required for capacitation. When the coimmunoprecipitation assay was performed, using capacitated spermatozoa in the presence of calpeptin, NOX2 co-immunoprecipitated with NOX4, and these results suggest that calpain regulates the dissociation of NOX2-NOX4 (Figure 6A,B).

Analysis of the protein recovered after incubation with the agarose-Protein A/G-IgG anti-NOX4 beads supports the coimmunoprecipitation results. The Wb for NOX4 shows that most of the NOX4 in the sperm protein extract was associated with the beads in all assays (Figure 6B, second panel). In contrast, the Wb for NOX2 shows that this NOX was only retained on the beads in the assays of non-capacitated and capacitated in the presence of calpeptin (Figure 6B, third panel).

### 2.8. Interrupting the Separation of NOX2-NOX4 Prevents the Production of H_2_O_2_

Once we determined that calpains mediate the separation of NOX2-NOX4 during capacitation, we investigated whether the production of H_2_O_2_ by both NOXs depends on this separation process. We assessed H_2_O_2_ production during capacitation when calpain was inhibited by calpeptin. Figure 7 shows that H_2_O_2_ production significantly increases during capacitation (Figure 7A). This increase was significantly inhibited when the spermatozoa were capacitated in the presence of calpeptin (Figure 7A).

To determine whether calpain inhibition resulted in capacitation, PYP was assessed in capacitated spermatozoa in the absence and presence of calpeptin (10 µM). The results showed a significant increase in PYP in capacitated spermatozoa compared to non-capacitated spermatozoa (Figure 7B,C). When the spermatozoa were capacitated in the presence of calpeptin, an increase in PYP did not occur (Figure 7C). The previous results suggest that calpain inhibition during capacitation prevents ROS production and PYP, with both processes related to capacitation.

## 3. Discussion

ROS production in mammalian spermatozoa is a consequence of the different signaling pathways in these cells. ROS activate other signaling pathways related to processes such as capacitation, acrosomal reaction, and motility [1,4,21]. Despite the physiological and pathological evidence of the effects of ROS on spermatozoa, the identity of the NADPH oxidases responsible for ROS production remains uncertain, although two different NOXs, NOX2 and NOX5, are present in mammalian spermatozoa [7,9,23,24,25]. The present work demonstrates clear evidence of the presence of two different NOXs in mouse and guinea pig spermatozoa, NOX2 and NOX4, which increase the levels of O_2_^•^- and H_2_O_2_ during capacitation. The evidence presented here directly relates NOX2 and NOX4 to sperm processes, such as capacitation, spontaneous acrosomal reaction (sAR), and motility. Notably, our results reveal an unprecedented mechanism related to NOX family activation, specifically NOX2 and NOX4, which occurs during capacitation and involves the Ca^2+^-dependent protease calpain.

Different reports suggest that ROS production in mammalian spermatozoa could be the consequence of a single member of the NOX family. NOX5 occurs in human, stallion, and canine spermatozoa [7,22,23,26], and NOX2 in mouse, rat and goat spermatozoa [8,9,25]. This suggestion is supported by the study by Musset et al., which demonstrated that human spermatozoa do not express NOX1, NOX2, or NOX4 [7]. Our results show that, as in other rodents [24,25], NOX2 is also expressed in guinea pig spermatozoa (Figure 1B). At the same time, analysis of the consumption of NADPH revealed the presence of another member of the NOX family in guinea pig spermatozoa, NOX4. ROS production by NOX2 requires active Rac1 [3]; therefore, the increase in NADPH consumption that occurs during capacitation (Figure 1A) was not suppressed by a specific Rac1 inhibitor (NSC23766). In contrast, NADPH consumption increased (Figure 1A); this is indicative of the presence of other NOXs not regulated by Rac1, and resulted in NOX4, which was corroborated by WB (Figure 1B) and immunolocalization (Figure 6A). It is possible that spermatozoa express not only NOX2 and NOX4 but also NOX1 and NOX3. Nevertheless, this remains to be clarified in a later study.

Our results show the presence of two different NOXs in guinea pigs and mouse spermatozoa, but further evidence also suggests that both NOX2 and NOX4 are active in guinea pig spermatozoa. In the case of NOX2, we know that the main product of this enzyme is the O_2_^•^-, which is transmuted to H_2_O_2_ [2]. NSC23766 and VAS2870 inhibit superoxide ion production at levels found in non-capacitated spermatozoa (Figure 2B). Thus, the production of the O_2_^•^- is blocked by inhibiting Rac1, indicating that this anion is produced by a NOX dependent on Rac1, such as NOX2. However, Rac1 inhibition did not reduce H_2_O_2_ levels; these were reduced by ML171 (Figure 2C), a more specific inhibitor for NOX4 that does not affect NOX2 activity [27]. Therefore, we suggest that the production of H_2_O_2_ under this condition must be due to a NOX not dependent on Rac1, such as NOX4. The fact that NOX2 inhibition does not reduce H_2_O_2_ levels during capacitation does not rule out the transmutation from O_2_^•^- to H_2_O_2_ under normal conditions. It is essential to consider that the reduction in H_2_O_2_ levels compared to those observed for non-capacitated spermatozoa is only achieved when NSC23766 and ML171 were tested together (Figure 2C). In conclusion, our data clearly show that NOX2 and NOX4 are responsible for ROS production during capacitation.

The use of ML171 (2 µM) in this work implies the possibility that NOX1, NOX4, and NOX5 activity may be affected [27]. First, it is known that NOX5 is an NADPH oxidase exclusive to human beings [7]. Since the gene does not exist in organisms of the rodent order [2], there is no interference in the production of ROS in the spermatozoa of rodents such as guinea pigs and mice. In the case of NOX1, ML171 concentrations less than 0.5 µM specifically inhibit NOX1 [28]. However, at these concentrations, the production of H_2_O_2_ did not show changes concerning capacitated spermatozoa in the absence of ML171. NOX1, if present in rodent spermatozoa, would not be involved in the different physiological processes experienced by the sperm before fertilizing the egg. This is supported by the fact that NOX1-deficient mice are fertile [28]. Finally, the NOX2 activity could not have been altered by ML171, since it is only affected at concentrations greater than 10 µM [27,28].

During capacitation, O_2_^•^- and H_2_O_2_ levels rose due to the increased activity of the enzymes responsible for producing these ROS [1,4,29]. Several enzyme pathways responsible for ROS production and their participation in processes such as capacitation, acrosomal reaction, and motility have been characterized [29]. However, except for NOX5 [7], little is known about the participation of NOX2 and NOX4 in ROS production and its impact on sperm physiology. Our results clearly show that NOX2 and NOX4 inhibition by an agonist of both NOXs and VAS2879 [27] almost entirely prevents the production of O_2_^•^- and reduced the levels of H_2_O_2_ (Figure 2B,C). However, this low level of H_2_O_2_ production does not support capacitation (Figure 4 and Figure 5) and directly affects sperm processes such as acrosomal reaction and the activation of progressive and hyperactivated motility. Therefore, the inhibition of O_2_^•^- and H_2_O_2_ production by VAS2870 during capacitation directly relates NOX2 and NOX4 to physiological processes such as capacitation and motility, which require an adequate level of O_2_^•^- and H_2_O_2_ produced by NOX2 and NOX4 for these sperm processes to occur correctly. In this regard, it is well known that O_2_^•^- and H_2_O_2_ are related to capacitation, acrosomal reaction, and motility by activating different kinases, such as PKA, PKC, and MEK, which are fully involved in sperm physiology [29,30].

The early increase in the B and AR patterns experienced by sperm when VAS2870 inhibits H_2_O_2_ production suggests that spermatozoa undergo early capacitation; therefore, sAR is also increased early (Figure 3A,B). These data appear to be contradictory regarding the non-increase in PYP during capacitation in the presence of VAS2870, and suggest that the spermatozoa would not undergo capacitation (Figure 3C,D). The development of the B and AR patterns through the CTC assay is dependent on Ca^2+^, since CTC interacts with membrane proteins in the presence of this cation [31]. While the adequate production of H_2_O_2_ during capacitation is related to the activation of different kinases, such as PKC and PKA, these kinases show low activity when capacitation is performed in an environment with low levels of H_2_O_2_ [29]. We suggest that both processes, the association of CTC with membrane proteins and PYP, occur independently during capacitation, since, despite the low production of H_2_O_2_ when NOX2 and NOX4 are inhibited, a continuous sperm influx Ca^2+^ (Figure 5) and, therefore, the association between CTCs and membrane proteins, should occur during capacitation. In this sense, a recent study reported that in human sperm, during capacitation, progesterone increases Ca^2+^ influx and ROS production by NOX5. Still, the inhibition of NOX5 by DPI, in both the presence and absence of progesterone, reduces the production of ROS but not the influx of Ca^2+^ [32]. We also suggest that normal H_2_O_2_ production is required for capacitation and AR events to occur normally. Low H_2_O_2_ production during capacitation induces a higher Ca^2+^ uptake, leading to the early occurrence of these events.

The presented results reveal a novel mechanism for the activation of NOXs, in which calpain-1, the calpain expressed in guinea pig spermatozoa [14], plays a predominant role. In spermatozoa in a non-capacitated state, NOX2 and NOX4 are physically associated. In this physiological state, the production of O_2_^•^- and H_2_O_2_ is deficient (Figure 3 and Figure 7). Therefore, we suggest that the interaction between NOX2 and NOX4 keeps these two NADPH oxidases inactive or at very low activity levels (Figure 7A). The interaction between NOX2 and NOX4 changes when spermatozoa enter the capacitated state, separating and increasing their activity (Figure 3 and Figure 7). NOX2-NOX4 interaction occurs independently of their activity since, even in the presence of VAS2870, the NOXs are separated (Figure 6B). Additionally, as H_2_O_2_ production increases only in the sperm-capacitated state (Figure 3 and Figure 7), we propose that the separation of NOX2 and NOX4 is a crucial point in the activation of these two NOXs, and also suggest that the increase in ROS causes capacitation and hypermotility to occur without alterations.

Furthermore, calpain-1 is known for its importance in sperm processes, including capacitation, acrosomal reaction, and motility, in different mammals’ spermatozoa [12,13,14,16,18]. Although different relationships between NOXs and calpain have been reported, wherein NOX2 and NOX4 regulate calpain activity in neuropathological processes [10,11,33], a mechanism where calpain-1 is related to the activation of NOX2 and NOX4 has not been previously reported. Our results suggest that calpain-1 activation is an essential and critical requirement for capacitation and motility to occur through the separation of NOX2 and NOX4, leading to an increase in O_2_^•^- and H_2_O_2_, which are directly related to capacitation and sperm motility [1,4,29].

We do not know whether the NOX2/NOX4 interaction is direct or whether other proteins are involved. The relationship between NOX2 and NOX5 and caveolin-1 has been characterized, and it has been suggested that caveolin-1 is a negative regulator of these two NOXs [34,35]. Additionally, the caveolin-1/NOX2 interaction was observed in goat spermatozoa, and NOX2 activity is closely related to the reorganization of lipid rafts that occurs during capacitation [9]. The relocation experienced by NOX2 and NOX4 during capacitation (Figure 7A) is similar to that experienced by lipid rafts in guinea pig spermatozoa [15]. Although neither caveolins nor flotillins are targeted by calpains, we suggest that these proteins, which are related to lipid rafts, could be associated with the reorganizing of these two NOXs that occurs after their separation. Recently, in cardiomyocytes, it was reported that the NOX2 and NOX4 activity depends on their interaction with LRRC8A through its LRRD domain [34]. However, like lipid raft proteins, LRRC8A is not targeted by calpain. It is important to note that, in the work of Huo et al. (2021), as in this work, the involved NADPH oxidases are NOX2 and NOX4, suggesting that these two NOXs could have an important relationship in multiple cellular and physiological processes. Therefore, it remains to be elucidated whether the NOX2/NOX4 interaction is direct or mediated by other proteins.

In conclusion, our results indicate that the production of O_2_^•^- and H_2_O_2_ during capacitation is related to at least two NOXs: NOX2 and NOX4. The inhibition of these two NOXs implies a low production of O_2_^•^- and H_2_O_2_ during capacitation and, as a result, capacitation, acrosome reaction and motility are affected, directly relating these two NOXs with the physiological processes that enable spermatozoa to fertilize. Therefore, the normal production of ROS by NOX2 and NOX4 during capacitation is essential for events such as capacitation, acrosomal reaction, and motility to occur normally. Finally, we present a new regulatory mechanism for NOXs, directed by the calcium-dependent protease calpain.

## 4. Materials and Methods

### 4.1. Reagents

All reagents used in this work were obtained from Sigma-Aldrich (St. Louis, MO, USA), except where otherwise indicated.

### 4.2. Experimental Animals

Male Dunkin-Hartley guinea pigs (*Cavia porcellus*) with a mean weight from 800 to 900 g were used to isolate testicles, epididymis, and vas deferens. Experimental animals were handled in accordance with the Mexican Official Norm for Laboratory Animals (NOM-062-ZOO-1999) and protocols from the Internal Committee for the Care and Use of Laboratory Animals, Cinvestav-IPN (CICUAL No. 321-02), following American Veterinary Medical Association guidelines.

### 4.3. Capacitation Assay

Spermatozoa were obtained from the vas deferens of guinea pigs in 2 mL of phosphate-buffered saline (PBS, pH 7.4) perfusion. The cells were adjusted to 3.5 × 10^7^ cells/mL and incubated at 37 °C during and until capacitation. We used minimal capacitate medium supplemented with sodium pyruvate and lactic acid (MCM-PL (NaCl 105.8 mM, CaCl_2_ 1.8 mM, NaHCO3 25.1 mM, lactic acid 20 mM, sodium pyruvate 0.25 mM and HEPES 2.5 mM, pH 7.8)) to induce capacitation. For non-capacitive conditions, the cell suspension (3.5 × 10^7^ cell/mL) was incubated in MCM-PL lacking CaCl_2_ and NaHCO_3_ at 37 °C.

### 4.4. Pharmacologic Inhibitors

ROS production in guinea pig spermatozoa was assessed under non-capacitive and capacitive conditions, with or without specific NOX inhibitor, VAS2879 (40 µM), a pan-NOX inhibitor [27]; ML171 (2 µM) or NSC23766, a small GTPase Rac1 inhibitor (100 µM). We also tested the effect of calpain inhibition on ROS production using calpeptin (10 µM), a calpain-specific inhibitor.

### 4.5. Assessment of NADPH Consumption

NADPH consumption was assessed following the method described by Andrade-Pavon et al. [36]. Whole native protein extracts obtained from spermatozoa under non-capacitation and capacitation conditions were tested with or without specific inhibitors of NOXs in the presence of NADPH (18 nM). The decrease in absorbance at 340 nm was measured for 10 min, and the difference between the initial and final absorbance was used to calculate the enzymatic activity (µmol/min/mg–protein)**.**

### 4.6. Assessment of Oxidative Stress

Intracellular peroxide assay. Intracellular ROS levels were determined using the oxidation-sensitive fluorescent probe 2′,7′-dichlorodihydrofluorescein diacetate (H2DCF-DA) to analyze the intracellular content of peroxides (H_2_O_2_) following a procedure modified from Guthrie and Welch [37]. Spermatozoa (3.5 × 10^7^ cell/mL) were incubated under conditions of capacitation in the presence of H2DCF-DA (50 µM final concentration), at 37 °C for 60 min in the dark. ROS production was evaluated (Ex/Em, 490/525) using an LS55 Luminescence Spectrometer (Perkin Elmer, Waltham Mass. USA). Detection of extracellular H_2_O_2_ by Amplex UltraRed (AUR) assay in the presence of HRP; spermatozoa (3.5 × 10^7^ cell/mL) were incubated under conditions of capacitation with or without specific inhibitor in the presence of AUR (50 µM final concentration) and HRP (0.1 mM final concentration), at 37 °C for 60 min in the dark. Samples were centrifuged (5000× *g*) for 3 min and supernatants were placed in a 96-well plate. Absorbance was read on a Microplate Reader at 540 nm (Bio rad, microplate reader, model 550).

Superoxide anion assay: Nitrotetrazolium Blue (NBT, 0.6 mM) was used to determine O_2_^•^- [38]. ROS production assay was performed at 3.5 × 10^7^ cell/mL during capacitation (60 min). The NBT deposited inside the cells was then dissolved by adding KOH (2 M) to solubilize cell membranes and then adding DMSO to dissolve Formazan complex by gentle shaking for 10 min at room temperature. Absorbance was measured at 620 nm. Relative production was obtained when sperm were incubated in PBS, relative solution was then transferred to a 96-well plate and absorbance was read on a Multi-Function Microplate Reader at 620 nm (Power Wave X 340, Bio-Tek Instruments, Winooski, VT, USA).

### 4.7. Effect of VAS2870, NSC23766, and DPI on Sperm Viability

Sperm viability was determined following the method described by Roa-Espitia et al. [39]. Sperm suspensions were incubated in MCM-PL at pH 7.8 and 37 °C for 60 min in the absence and presence of VAS2870 (40 µM) or NSC23766 (100 µM). Once the incubation was finished, we added a PI solution (1 μg/mL) to a sample of spermatozoa to a 1:1 ratio and mixed and incubated the mixture at room temperature for 30 min. The spermatozoa were washed and the number of stained and unstained spermatozoa was counted (500 cell X sample, *n* = 3) under an epifluorescence microscope (Olympus BX500, Tokyo, Japan). Images were registered and analyzed using the software Nikon Elements 3.1.

### 4.8. Western Blotting

Vas deferens sperm were obtained, washed in 154 mM NaCl solution, and capacitated. Samples were centrifuged (5000× *g*) for 3 min and suspended in RIPA buffer (25 mM TRIS HCl pH 7.6, 150 mM NaCl, 1% NP-40, 1%, sodium deoxycholate, 0.1% SDS), supplemented with protease inhibitors (5 mg/mL soybean trypsin inhibitor, 100 mg/mL benzamidine, 30 mg/mL pepstatin A, 30 mg/mL leupeptin, 30 mg/mL aprotinin, 1 mM PMSF diluted in dimethyl sulfoxide, 20 mg/mL iodoacetamide, 1 mM sodium orthovanadate, 10 mM sodium fluoride, 10% glycerol, and 2.5% complete, Mini, EDTA-free Protease Inhibitor Cocktail [1 tablet diluted in 1 mL H2O]). The samples were then incubated for 20 min on ice and centrifuged (20,000× *g*) for 20 min at 4 °C. The supernatants were collected, and their protein concentration was determined [40]. To reduce protein disulfide bonds, supernatant aliquots were boiled for 5 min in 3X Laemmli buffer (720 mM TRIS-base, 6% SDS (*w*/*v*), 30% glycerol, 2-mercaptoethanol, and 0.03% bromophenol blue (*w*/*v*), pH 10). Then, proteins were separated using sodium dodecyl sulfate-polyacrylamide gel electrophoresis (SDS-PAGE) on 10% polyacrylamide non-gradient gels, and subsequently transferred to nitrocellulose membranes and blocked in 5% skimmed milk in 1 × PBS/1% Triton X-100 (pH 7.4).

Phospho-tyrosine detection. First, 500 µg of total spermatic protein extract was run on SDS-PAGE gels and transferred to nitrocellulose membranes overnight at 4 °C. Membranes were washed once with PBS-Tween 20 (0.1%) and blocked with 5% nonfat dry milk in PBS-Tween 20 (0.1%) for two hours at room temperature with constant agitation. Membranes were then incubated with a 1:500 dilution of anti-phosphotyrosine (Cell Signaling, APY03, Danvers, MA, USA) antibody diluted in PBS-Tween 20 (0.1%) and incubated overnight at 4 °C with constant agitation followed by washing three times in PBS-Tween 20 (0.1% final concentration). The secondary mouse antibody HRP-conjugated IgG (Jackson Labs., 115-035-068) was diluted 1:10,000 in PBS added with nonfat milk (3% final concentration) and incubated with membranes for one hour at room temperature. Membranes were then washed three times in PBS for 10 min each.

Nox4 and Nox2 detection. Using SDS-PAGE, 300 µg of spermatic protein extract was loaded, and the proteins were separated and transferred to nitrocellulose membranes. Western blotting was performed under the same conditions used for phospho-tyrosine detection. The antibodies used were anti-NOX2/gp91^phox^ antibody (EPR6991) (Abcam, ab129068) for NOX2 detection and anti-NADPH oxidase 4 antibody (UOTR1B492) (Abcam, ab109225, Cambridge, UK) for NOX4 detection.

In both assays, proteins were detected by chemiluminescence, which was developed using Amersham Biosciences ECL Prime Western Blotting Detection Reagent and visualized in an Odyssey^®^ Fc Imaging System by LI-COR Biosciences. The densitometric analysis was performed using the ImageJ software (https://imagej.nih.gov (accessed on 3 April 2021)).

### 4.9. Chlortetracycline Fluorescence Assay

This procedure was previously described [31] and modified for guinea pig spermatozoa [39,41] as follows: The stain solution was prepared by dissolving 250-μM chlortetracycline (CTC)-HCl in TN buffer (20 mM Tris, 130 mM NaCl, and 5 mM cysteine at pH 7.8); fresh CTC stock was prepared daily. At the time of the assay, 20 μL of non-capacitated or capacitated spermatozoa, treated or not with VAS2879, were mixed with 20 μL of pre-warmed CTC stock solution and incubated for 20 s in a water bath at 37 °C. Immediately after incubation, the CTC-sperm suspension was fixed adding 3.5 µL of 12.5% glutaraldehyde in 1.25 M Tris (pH 7.5), immediately followed by gentle mixing. Fixed samples were kept in a dark box. After 1–4 h of fixation, slides were prepared and examined under fluorescence microscope (Ex330-380/Em420 nm). All fluorescence images were obtained using an Olympus BX500 fluorescence microscope. In each sample, 100 spermatozoa were classified as expressing one of three CTC staining patterns: F pattern, a faint fluorescence in the acrosome region, which is characteristic of non-capacitated acrosome-intact cells; B pattern, a bright fluorescence in the acrosomal region with a band along the equatorial segment, which is typical of capacitated, acrosome-intact cells; AR pattern, a fluorescence in the equatorial segment and/or post-acrosomal region, which is characteristic of physiologically capacitated acrosome-reacted cells. The presence or absence of the acrosomal cap on each cell was verified using phase-contrast illumination.

### 4.10. Spermatic Motility Analysis

Sperm motility was evaluated following the methodology described by Cordero-Martínez [41]. Guinea pig spermatozoa were capacitated for 60 min in the absence or presence of VAS2870 (40 µM). During incubation, aliquots were used for sperm motility quantitative parameter screening on a CASA instrument (TOX IVOS, software version 12.3: Hamilton Thorne Bioscience, Beverly, MA, USA) in a sample counting chamber prewarmed to 37 °C (MicroCell 20 Micron). Sperm motility was recorded at 60 frames/s for one second; velocity distribution and other kinematic parameters of motility (VAP, VSL, VCL, ALH, and BCF) were analyzed. Experiments were performed in triplicate.

### 4.11. Intracellular Calcium Evaluation

Spermatozoa were capacitated for 15, 30, 60 and 90 min in the absence or presence of VAS2870 (40 µM), and then spermatozoa (aliquots of 3.5 × 10^7^ cell/mL) were loaded with the Ca^2+^ indicator dye Fluo-3AM (Fluo-3-pentaacetoxymethyl ester, Invitrogen, Carlsbad, CA, USA) resuspended in dimethyl sulfoxide (DMSO, 1 mM). Assays were prepared at a final concentration of 2 μM Fluo-3AM for 15 min at 37 °C in MCM-PL. The cells were washed by centrifugation at 5000× *g* with MCM-PL. After treatment, changes in intracellular calcium concentration ([Ca^2+^]i) were recorded as background substrate ratios of the corresponding excitation wavelength (Ex490/Em520 nm). Intracellular calcium evaluation was performed for 200 s on a continuous basis, with or without pharmacological inhibitors. Fluorescence was registered in an LS-55 Fluorescence Spectrometer (Perkin Elmer, Waltham MA, USA).

### 4.12. Immunofluorescence Assays

Non-capacitated or capacitated spermatozoa, treated or not with calpeptin, were fixed in formaldehyde (1.5% final concentration) in PBS. After one hour, the sperm were collected by centrifugation. The pelleted sperm (600 g for 3 min) were incubated in 50 mM NH_4_Cl for 10 min, rinsed twice with PBS, and then with bidistilled water. Microscope slides were prepared using this suspension, air-dried at room temperature overnight and stored at 4 °C. Sperm cells were permeabilized in acetone for 7 min at −20°C and washed with PBS. NOX2 or NOX4 (1:100) antibody was diluted in PBS with 1% BSA (blocking solution) added and incubated overnight at room temperature. The slides were washed with PBS and incubated for two hours at 37 °C with the appropriate TRITC-conjugated secondary antibody diluted in blocking solution. The samples were mounted on glass-covered slides using Gelvatol, sealed properly, and stored at −20 °C until observations were performed. The stained cells were imaged under a confocal laser scanning microscope (Leica TCS SP8, Wetzlar, Germany) and analyzed using LAS AF Lite’s imaging software (Version 2.6.3).

### 4.13. Co-Immunoprecipitation Assay

Co-immunoprecipitation experiments were performed using the Crosslink Immunoprecipitation Kit (Thermo Fisher Scientific, Waltham MA, USA, 26147). This method involves capturing 20 μg antibody on Protein A/G beaded agarose resin and covalently immobilizing it on the support by crosslinking with 2.5 mM disuccinimidyl suberate (DSS). The antibody resin was then incubated at 4 °C for 12 h with 500 μg precleared guinea pig spermatozoa protein extracts, allowing for the antibody-antigen complex to form. Proteins bound to the respective antibodies were eluted, recovered by low-speed centrifugation (3000× *g*), and stored at −20 °C. Only the antigen was eluted during the procedure, enabling it to be identified and analyzed with minimal interference from antibody fragments. For protein disulfide reduction, supernatant aliquots were boiled for 5 min in 3X Laemmli buffer (pH 10) containing 2-mercaptoethanol, and the proteins were then separated by SDS-PAGE. Next, proteins were transferred to nitrocellulose membranes for immunodetection. The recovered proteins were analyzed by WB using the adequate antibodies.

Negative control was performed by associating 20 µg of an IgG unrelated to NOX2 or NOX4 with the agarose-protein A/G beads. The beads were incubated with 500 µg of sperm protein extract in the same way as before. The WB carried out with the anti-NOX2 antibody did not show the presence of this protein, nor of another (Appendix A).

### 4.14. Statistical Analysis

SigmaPlot 11 was employed to perform statistical analysis. Data are expressed as mean ± S.E. Means were compared using ANOVA one way as appropriate. Statistical significance between the samples was considered when *p* ≤ 0.05.

## Figures and Tables

**Figure 1 ijms-24-03980-f001:**
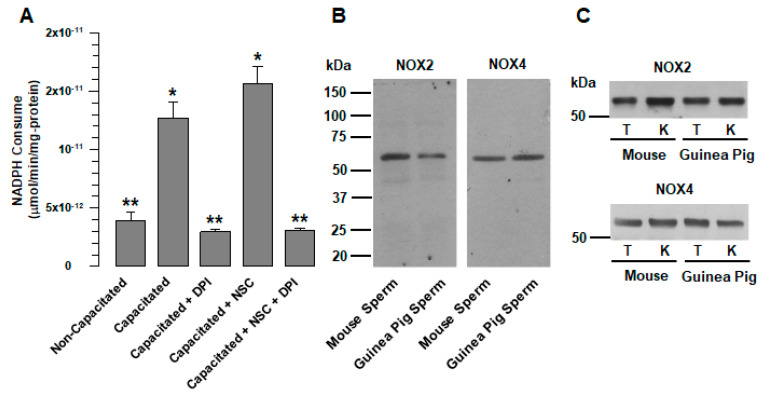
Guinea pig and mouse spermatozoa express two different NOXs. (**A**) Consumption of NADPH in total native protein extracts of guinea pig spermatozoa. The consumption of NADPH was assessed in both non-capacitated and capacitated spermatozoa in the absence and presence of DPI (10 µM) and NSC (NSC23766 100 µM). The results are expressed as the ratio N/N0, where N is the total amount of NADPH that is consumed and N0 is the total quantity of NADPH that is consumed in the non-capacitated assay. The results are expressed as the mean ± S.E. (*n* = 3). Differences were considered significant when *p* ≤ 0.05, comparing (*) concerning (**). (**B**) Detection of NOX2 and NOX4 in guinea pig and mouse spermatozoa. Proteins from whole spermatozoa extracts (300 µg) were resolved by SDS-PAGE and transferred to nitrocellulose membranes. The proteins were detected using specific antibodies against NOX2 and NOX4. MS: Mouse spermatozoa. GPS: Guinea pig spermatozoa. (**C**) Detection of NOX2 and NOX4 in guinea pig and mouse testes and kidneys. Proteins from whole extracts of guinea pig and mouse testes and kidneys were used to determine antibody specificity. T: Testis. K: kidney. Images are representative of three independent experiments. The representative Wb images (**B**,**C**) of three independent experiments for NOXs detection are shown.

**Figure 2 ijms-24-03980-f002:**
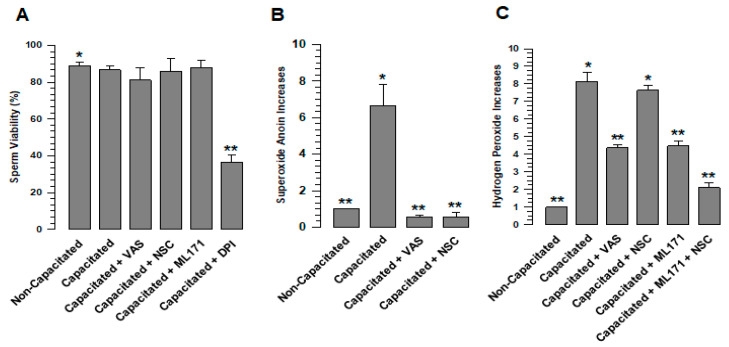
ROS production in sperm increases during capacitation. The status of ROS production was determined in guinea pig spermatozoa; they were capacitated for 60 min in the absence and presence of VAS (VAS2870 40 µM) or NSC (NSC23766 100 µM) or/and ML171 (2 µM), then O_2_^•^- and H_2_O_2_ were assessed. (**A**) VAS2870, NSC23766, and ML171 did not alter sperm viability. (**B**) Increase in O_2_^•^- production during capacitation. (**C**) Increase in H_2_0_2_ production during capacitation. The results are expressed in the ratio N/N0, where N is the total amount of O_2_^•^- or H_2_O_2_ produced in each assay and N0 is the total quantity of O_2_^•^- or H_2_O_2_ produced by non-capacitated assay. Results are expressed as the mean ± S.E. (*n* = 3). Differences were considered significant when *p* ≤ 0.05, comparing (*) concerning (**).

**Figure 3 ijms-24-03980-f003:**
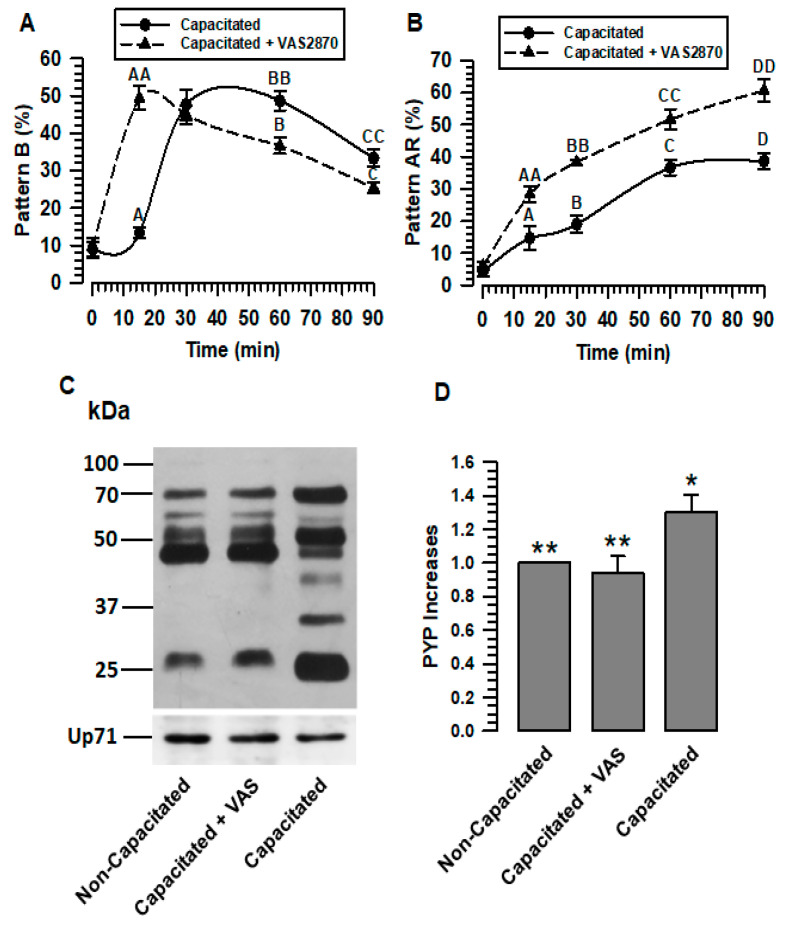
The activity of sperm NOXs is important for capacitation and acrosomal reactions. The role of NOX in guinea pig sperm physiology was defined by incubating them under capacitating conditions for 60 min, either in the absence or presence of VAS (VAS2870 40 µM). Capacitation was evaluated by CTC techniques, and protein tyrosine phosphorylation patterns (PYP) were determined by immunoblotting using a specific anti-p-Y antibody. (**A**) Assessment of pattern B by CTC. (**B**) Evaluation of the A.R. pattern by CTC. (**C**) Determination of PYP increases by immunoblotting in non-capacitated or capacitated spermatozoa in the absence and presence of VAS (VAS2870 40 µM). Up71 was used as a loading control. The representative Wb image of three independent experiments for PYP is shown. (**D**) Densitometric assessment of PYP. The results are expressed as the ratio N/N0, where N is the total amount of PYP and N0 is the total PYP quantity of non-capacitated. Results are expressed as the mean ± S.E. (*n* = 3). Differences were considered significant when *p* ≤ 0.05, comparing (*) concerning (**).

**Figure 4 ijms-24-03980-f004:**
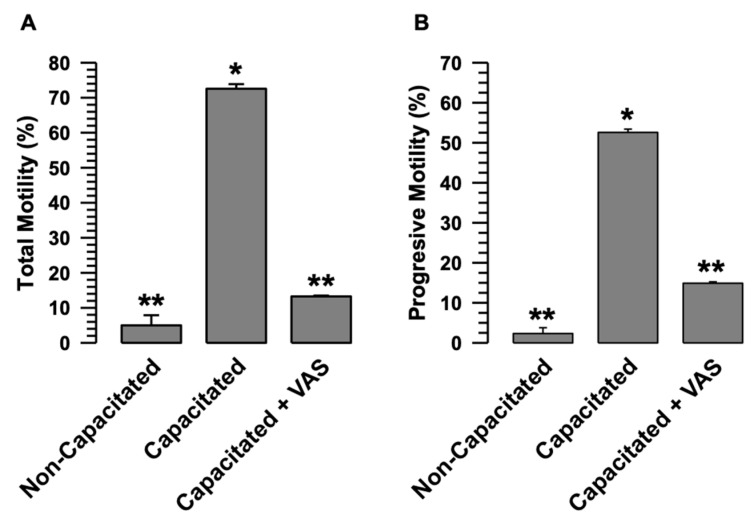
The activity of sperm NOXs is relevant to sperm motility. The role of NOXs in motility was determined; guinea pig spermatozoa were incubated under capacitive conditions in the absence and presence of VAS (VAS2870 40 µM). Sperm motility was assessed by CASA. (**A**) Total motility. (**B**) Progressive motility. Results are expressed as the mean ± S.E. (*n* = 3). Differences were considered significant when *p* ≤ 0.05, comparing (*) concerning (**).

**Figure 5 ijms-24-03980-f005:**
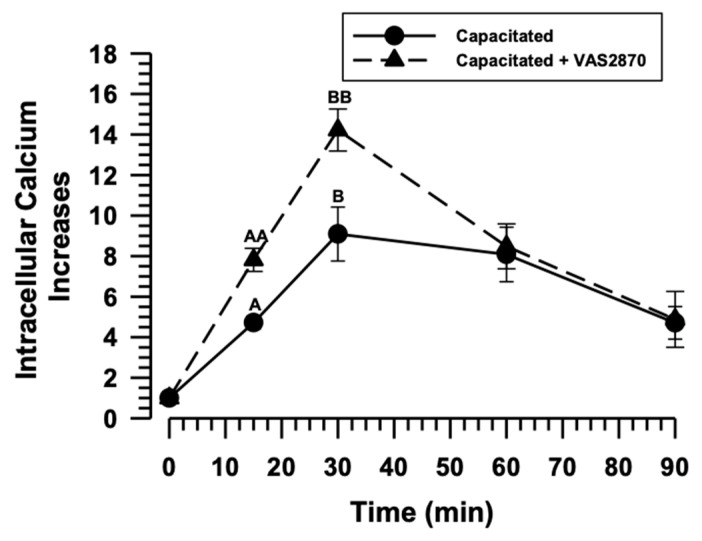
NOX inhibition during capacitation does not affect Ca^2+^ influx. The role of NOXs Ca^2+^ influx during capacitation was determined in guinea pig spermatozoa. The sperm were incubated under capacitation conditions in the absence and presence of VAS2870 (40 µM). The [Ca^2+^]i was recorded at different incubation times by spectrofluorimetry using Fluo-3AM. Results are expressed as the mean ± S.E. (*n* = 3). Differences were considered significant when *p* ≤ 0.05, comparing (A or B) concerning (AA or BB).

**Figure 6 ijms-24-03980-f006:**
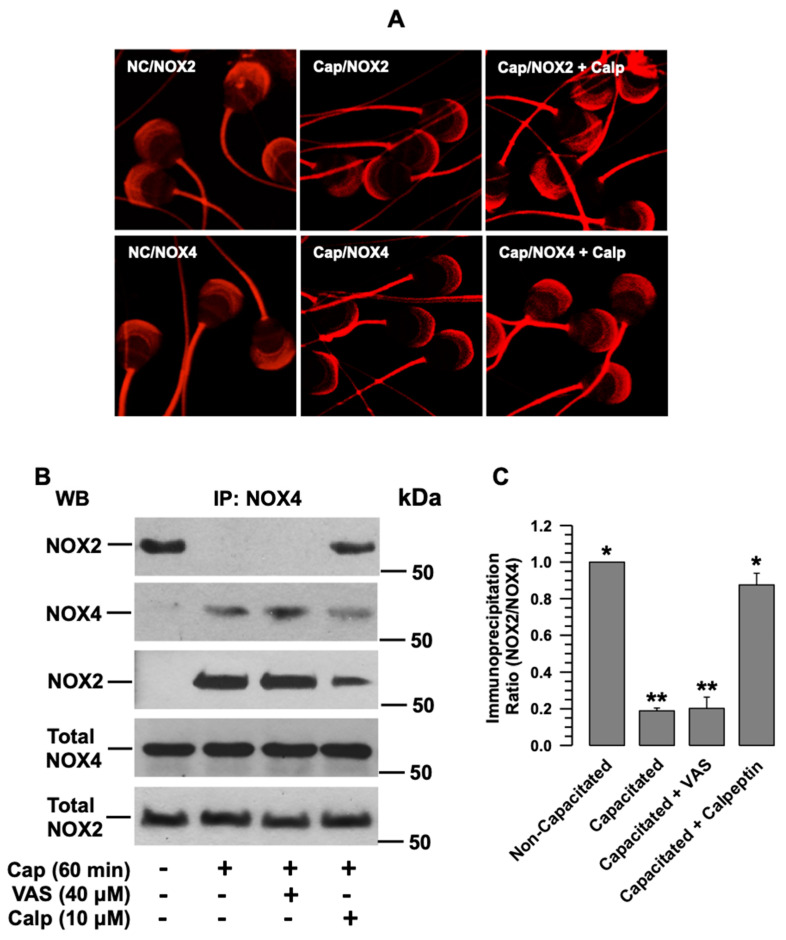
The interaction between NOX2 and NOX4 is broken during capacitation and depends on calpain activity. (**A**) Localization of NOX2 and NOX4 in non-capacitated (N.C.) and capacitated (Cap) sperm, either in the absence or presence of calpeptin (Calp, 10 µM). The images are representative of three independent experiments. (**B**) Coimmunoprecipitation of NOX2 with NOX4 in non-capacitated and capacitated (Cap) sperm in the absence and presence of VAS2870 (VAS) or calpeptin (Calp). The representative Wb image of three independent experiments for the co-immunoprecipitation assay is shown. (**C**) Densitometry of NOX2 coimmunoprecipitated with NOX4. The results are expressed as the ratio N/N0, where N is the total amount of NOX2 coimmunoprecipitated and N0 is the total quantity of NOX4 of each assay. Then, data were normalized concerning the non-capacitated data. VAS: VAS2870. Calp: Calpeptin. Results are expressed as the mean ± S.E. (*n* = 3). Differences were considered significant when *p* ≤ 0.05, comparing (*) concerning (**).

**Figure 7 ijms-24-03980-f007:**
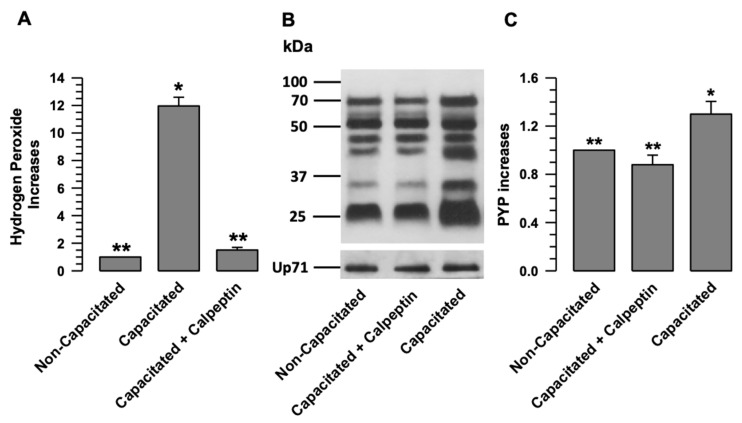
The activation of NOX2 and NOX4 depends on calpain activity. (**A**) H_2_O_2_ production was assessed in non-capacitated or capacitated sperm in the absence and presence of calpeptin (Calp 10 µM). The results are expressed as the ratio N/N0, where N is the total amount of H_2_O_2_ produced in each assay and N0 is the total quantity of H_2_O_2_ produced by non-capacitated assay. The results are expressed as the mean ± S.E. (*n* = 3). Differences were considered significant when *p* ≤ 0.05, comparing (*) concerning (**). (**B**) Determination of PYP patterns by immunoblotting of non-capacitated or capacitated sperm in the absence and presence of the calpain inhibitor calpeptin (Calp (10 µM)). Up71 was used as a loading control. The representative Wb image of three independent experiments for PYP is shown. (**C**) The results are expressed as the ratio N/N0, where N is the total amount of PYP and N0 is the total PYP quantity of non-capacitated assay. Results are expressed as the mean ± S.E. (*n* = 3). Differences were considered significant when *p* ≤ 0.05, comparing (*) concerning (**).

## Data Availability

Not applicable.

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
