# Peer review of "Calpain Regulates Reactive Oxygen Species Production during Capacitation through the Activation of NOX2 and NOX4"

_ijms, 2023, doi:10.3390/ijms24043980_

Round 1

Reviewer 1 Report

The abstract needs to be improved; it is difficult to understand

please remove the references from the result or make the result and discussion one part.

the figures of the result can be more clear

Author Response

Reviewer 1

First, we thank the reviewer for his valuable comments since they have improved our manuscript.

The abstract needs to be improved; it is difficult to understand.

Reply. The abstract was rewritten and revised for a better understanding (lanes: 18-38). The manuscript was edited in the English language by MDPI.

please remove the references from the result or make the result and discussion one part.

Reply. We followed the reviewer's suggestion, and the references in the results section were removed.

the figures of the result can be more clear

Reply. In order to make the figures more precise, they were redrawn, and some of the figure captions were improved.

Reviewer 2 Report

The manuscript entitled „Calpain regulates reactive oxygen species production during capacitation through the activation of NOX2 and NOX4” describes the role of mentioned proteins in spermatozoa capacitation. Sperm capacitation plays an important role in egg fertilization, which requires further investigation of the mechanisms of this process. Authors revealed the two enzymes, belonging to the NADPH oxidase (NOX) family, NOX2 and NOX4 in guinea pig spermatozoa. Further, the authors have demonstrated the participation of two enzymes in ROS production and spermatozoa capacitation. The authors have shown that NOX2 and NOX4 via calpain interaction participate in spermatozoa capacitation by ROS production.

I have a few comments:

1.      In lane 108, the first should be written in a lowercase letter.

2.      Fig 1: in part A there is no description of the two last columns, which compounds and conditions were used. Please correct. 

3.      I suggest moving part 2.2. „DPI affects sperm viability” afterward (or include in) point 2.3. It would be easier to read this part. In my opinion, part 2.2 is a consequence of the observation made in part 2.3, and would be logical to continue to study the viability of spermatozoa following the detection of ROS production and the role of NOX in spermatozoa capacitation.

4.      In point 2.3. there is no explanation of what VAS2870 inhibits. There is information about VAS2870 in the section „Materials and Methods”. I suggest introducing also the enzyme specificity of VAS2870 in the Result text.

5.      The image 3C and 7B represent WB analysis of PYP. There are a lot of bands. Could you, please, point to the right ones with an arrow? I understand that Fig 3C and 7C show only a representative image of three independent experiments. According to the description of Fig.3C and Fig 7C, there should be three images from three independent experiments, not only one. I suggest correcting the figure description. Could you describe which bands have been used for densitometric analysis?

Fig 6B shows the results of co-IP experiments. The image of the WB analysis of immunoprecipitates is very clear and nice. But there is a lack of input analysis and also negative control using IgG for immunoprecipitation. It would be desirable to include this information. It is also desirable to add lanes corresponding to molecular weight bands.  

Author Response

 Reviewer 2

Comments and Suggestions for Authors

The manuscript entitled „Calpain regulates reactive oxygen species production during capacitation through the activation of NOX2 and NOX4” describes the role of mentioned proteins in spermatozoa capacitation. Sperm capacitation plays an important role in egg fertilization, which requires further investigation of the mechanisms of this process. Authors revealed the two enzymes, belonging to the NADPH oxidase (NOX) family, NOX2 and NOX4 in guinea pig spermatozoa. Further, the authors have demonstrated the participation of two enzymes in ROS production and spermatozoa capacitation. The authors have shown that NOX2 and NOX4 via calpain interaction participate in spermatozoa capacitation by ROS production.

I have a few comments:

First, we thank the reviewer for his valuable comments since they have improved our manuscript.

  1. In lane 108, the first should be written in a lowercase letter.

 Reply. The English edition removed "First".

  1. Fig 1: in part A there is no description of the two last columns, which compounds and conditions were used. Please correct. 

Reply. The mistake has been fixed, and columns four and five of the figure 1A now show its description.

  1. I suggest moving part 2.2. „DPI affects sperm viability” afterward (or include in) point 2.3. It would be easier to read this part. In my opinion, part 2.2 is a consequence of the observation made in part 2.3, and would be logical to continue to study the viability of spermatozoa following the detection of ROS production and the role of NOX in spermatozoa capacitation.

Reply. For us, the reviewer's suggestion is important. However, DPI has been used in various works as an inhibitor of NADPH oxidases. Therefore, we consider it important to retain this section, which indicates that care must be taken with DPI because it drastically reduces sperm viability. In order to follow a logical presentation of the results, the viability plot is now presented in Figure 2A.

  1. In point 2.3. there is no explanation of what VAS2870 inhibits. There is information about VAS2870 in the section „Materials and Methods”. I suggest introducing also the enzyme specificity of VAS2870 in the Result text.

Reply. Considering the reviewer's suggestion, information about VAS2870 was introduced in point 2.3. (lanes: 149-151).

  1. The image 3C and 7B represent WB analysis of PYP. There are a lot of bands. Could you, please, point to the right ones with an arrow? I understand that Fig 3C and 7C show only a representative image of three independent experiments. According to the description of Fig.3C and Fig 7C, there should be three images from three independent experiments, not only one. I suggest correcting the figure description. Could you describe which bands have been used for densitometric analysis?

Reply. The error in figures 3C and 7B was corrected and now says: The image represents three independent experiments.

The densitometry was carried out using the ImageJ software that allows the densitometry of all the bands of a complete lane to be carried out together, obtaining a value for the entire lane. The use of this program for densitometry analysis is now indicated in the manuscript. The data provided by the program are normalized concerning the values obtained for non-capacitated spermatozoa. The procedure is standard for comparing protein phosphorylation that occurs during capacitation and is used by many laboratories. The use of this software is now indicated in the manuscript (lanes: 579-580).

 With respect to indicating each band with an arrow, it would make the figure more complex and could confuse the reader.

1  Fig 6B shows the results of co-IP experiments. The image of the WB analysis of immunoprecipitates is very clear and nice. But there is a lack of input analysis and also negative control using IgG for immunoprecipitation. It would be desirable to include this information. It is also desirable to add lanes corresponding to molecular weight bands.  

Reply. The inputs were analyzed by Wb; the results are described in the manuscript and presented in figure 6B (lanes: 289-294).

A negative control that we commonly perform in co-immunoprecipitation assays is to carry out the same procedure using a non-specific IgG and, by Wb determine that none of the tested proteins associates non-specifically with the agarose-protein A/G-IgG beads. Here we show the control carried out for the co-immunoprecipitation of NOX2 with NOX4. As can be seen, NOX4 is not associated with the column; therefore, NOX2 was not detected in the WB. Almost all NOX4 was recovered in the supernatant recovered after the incubation. This WB is not shown in the results, but the control is mentioned in the manuscript (lanes: 652-656, Supplemental figure 1).

Following the reviewer's suggestion, the corresponding molecular weights of the bands are now indicated in Figure 6B.

Round 2

Reviewer 1 Report

It is ok

Author Response

We appreciate the reviewer's valuable comments that have allowed us to improve this manuscript significantly.

The manuscript was sent for proofreading and spelling checking by MDPI.

Reviewer 2 Report

Dear Authors,

Thank you for the corrections made. They significantly improved the manuscript. However, I still have a few comments.

1.      In Fig. 2C, there is a lack of a last bar label. Please, fill the gap.

2.      I am also wondering whether the sentence” The image represents three independent experiments” is correct. I can interpret that the authors prepared protein extracts in three different experiments. Afterward, the protein extracts have been combined and have been used for Western blot only one time.  Is it true? In this case, when the authors performed Western blot (three times) using protein extracts obtained in three independent experiments,  it is correct to write „The representative western blot image of three independent experiments for PYP is shown”. Could you explain and correct this?

Author Response

We appreciate the reviewer's valuable comments that have allowed us to improve this manuscript significantly.

The manuscript was sent for proofreading and spelling checking to MDPI.

Reviewer 2

We appreciate the reviewer’s valuable comments that have allowed us to improve this manuscript substantially.

  1. In Fig. 2C, there is a lack of a last bar label. Please, fill the gap.

The description is now shown in the last column of Figure 2C.

  1. I am also wondering whether the sentence” The image represents three independent experiments” is correct. I can interpret that the authors prepared protein extracts in three different experiments. Afterward, the protein extracts have been combined and have been used for Western blot only one time.  Is it true? In this case, when the authors performed Western blot (three times) using protein extracts obtained in three independent experiments, it is correct to write „The representative western blot image of three independent experiments for PYP is shown”. Could you explain and correct this?

The phrase "The image represents three independent experiments" refers to the fact that this image is one of three independent experiments carried out and illustrates that similar results were obtained in each experiment. To avoid confusing the reader, the sentence “The image represents three independent experiments” was removed.  Taking into account the reviewer's recommendation, the phrase "The representative Wb image of three independent experiments for PYP is shown" was included in the text (lanes 216-217, 332-333).